# *tec-1* kinase negatively regulates regenerative neurogenesis in planarians

**Alexander Karge[1], Nicolle A Bonar[1], Scott Wood[1], Christian P Petersen[1,2]***

[1]Department of Molecular Biosciences, Northwestern University, Evanston, United States; [2]Robert Lurie Comprehensive Cancer Center, Northwestern University, Evanston, United States

**Abstract** Negative regulators of adult neurogenesis are of particular interest as targets to enhance neuronal repair, but few have yet been identified. Planarians can regenerate their entire CNS using pluripotent adult stem cells, and this process is robustly regulated to ensure that new neurons are produced in proper abundance. Using a high-throughput pipeline to quantify brain chemosensory neurons, we identify the conserved tyrosine kinase *tec-1* as a negative regulator of planarian neuronal regeneration. *tec-1*RNAi increased the abundance of several CNS and PNS neuron subtypes regenerated or maintained through homeostasis, without affecting body patterning or non-neural cells. Experiments using TUNEL, BrdU, progenitor labeling, and stem cell elimination during regeneration indicate *tec-1* limits the survival of newly differentiated neurons. In vertebrates, the Tec kinase family has been studied extensively for roles in immune function, and our results identify a novel role for *tec-1* as negative regulator of planarian adult neurogenesis.

## Introduction

The capacity for adult neural repair varies across animals and bears a relationship to the extent of adult neurogenesis in the absence of injury. Adult neural proliferation is absent in *C. elegans* and rare in *Drosophila*, leaving axon repair as a primary mechanism for healing damage to the CNS and PNS. Mammals undergo adult neurogenesis throughout life, but it is limited to particular brain regions and declines with age (*Tanaka and Ferretti, 2009*). By contrast, select organisms can undergo more extreme neural repair and also typically undergo more extensive ongoing neural differentiation through adulthood. Axolotls and zebrafish are able to repair extensive damage (*Tanaka and Ferretti, 2009*), while organisms such as planarians and hydra are able to regenerate an entirely new brain through whole-body regeneration (*Tanaka and Reddien, 2011*). Based on identification of similar mechanisms of regeneration from the distantly related planarians and acoels, whole body regeneration has been proposed to be an ancestral feature (*Gehrke and Srivastava, 2016*; *Reddien, 2018*). Understanding the regulatory mechanisms of adult neurogenesis could therefore reveal methods to enhance neural repair across species. In particular, specific negative regulators of neurogenesis would be important targets for repair enhancement, but few have yet been identified (*He and Jin, 2016*).

The planarian *Schmidtea mediterranea* undergoes complete brain regeneration within 1–2 weeks after injury and also perpetual homeostatic replacement of brain tissue, making this species an ideal model to identify such factors. The planarian brain is composed of an anterior bi-lobed cephalic ganglia with axon-rich neuropils containing interneurons and glia, as well as lateral branches with chemosensory, mechanosensory, and other neurons (*Agata et al., 1998*; *Mineta et al., 2003*; *Nakazawa et al., 2003*; *Roberts-Galbraith et al., 2016*; *Wang et al., 2016*). Two ventral nerve cords relay signals to the body through a peripheral nervous system, with many neuron subtypes, including serotonergic, GABAergic, dopaminergic, octopaminergic, cholinergic, and glutaminergic neurons (*Nishimura et al., 2007a*; *Nishimura et al., 2007b*; *Cebrià, 2008*; *Nishimura et al., 2008a*;

*For correspondence:
christian-p-petersen@
northwestern.edu

**Competing interests:** The authors declare that no competing interests exist.

*Nishimura et al., 2008b*; *Collins et al., 2010*; *Nishimura et al., 2010*). In addition, planarian cell atlas projects recently revealed the existence of ~50 distinct types of neurons, pointing to considerable complexity of regulation for producing and maintaining neurons through ongoing differentiation (*Fincher et al., 2018*; *Plass et al., 2018*). Planarian tissues are maintained and regenerated by *piwi-1+* neoblast stem cells, a mesenchymal cell population found in a parenchymal region and constituting the animal's only proliferative cells. Transcriptional profiling and candidate approaches have identified subpopulations of neoblasts specified for participation in distinct tissue lineages including a subpopulation of TSPN+ neoblasts that can give rise to all cell types (*Lapan and Reddien, 2011*; *Cowles et al., 2013*; *Cowles et al., 2014*; *Scimone et al., 2014*; *Brown et al., 2018*; *Zeng et al., 2018*). Neural progenitors have been identified as neoblast subpopulations expressing either proneurogenic transcription factors or transcription factors expressed in unique differentiated neurons, and some have been assigned to particular lineages through RNAi (*van Wolfswinkel et al., 2014*; *Molinaro and Pearson, 2016*; *Fincher et al., 2018*; *Plass et al., 2018*; *Ross et al., 2018*). Thus, neoblasts and their differentiating progeny sustain ongoing neurogenesis in the absence of injury as well as in regeneration of a new head.

Brain regeneration in planarians is a robust process that always ensures a proper restoration of relative neuron abundance in the animal. Decapitation triggers waves of wound-induced signals, bursts of proliferation, a patterning process to sense missing tissues, and differentiation of new tissues within an outgrowing blastema. Planarians do not grow appreciably as they regenerate, so the end of regeneration is characterized by a return to uninjured cell and tissue proportionality with respect to body size (*Oviedo et al., 2003*; *Hill and Petersen, 2015*). In addition, the regeneration process involves re-scaling and integrating pre-existing tissues with new tissues. For example, regenerating head fragments reduce the size of their brain while they form a small tail blastema until they reach appropriate proportionality. The robustness of the regeneration process suggests first that the system is well suited for identifying subtle phenotypes affecting adult neurogenesis. Secondly, the planarian's ability to attain and maintain a predictable relative abundance of neurons suggests the process is under both positive and negative control.

We report here the identification of a Tec non-receptor tyrosine kinase gene through RNAi screening in planarians that limits neuron abundance in regeneration and homeostatic tissue maintenance. Analysis of *tec-1*'s mechanism of action reveals it is unlikely to control the process of differentiation, but instead negatively regulates neuron survival. These results suggest that like neural development, both regeneration and stem-cell dependent adult neurogenesis both involve an initial overproduction of new neurons and identify a new factor involved in limiting the specific amplification of cells produced by adult neurogenesis.

## Results

### *tec-1* limits the regeneration of brain neurons

To identify negative regulators of neuronal regeneration in planarians, we conducted an RNAi screen of ~50 kinases and receptors with expression enriched in neoblasts (*Supplementary file 1*). In order to quantitatively measure effects on neuronal regeneration, we used a histological assay for staining and enumerating chemosensory neurons of the brain expressing the *cintillo* gene homologous to the Degenerin superfamily of sodium channels (*cto*, *Figure 1A*). *cto*-expressing neurons are arranged in a stereotyped pattern in close proximity to the lateral brain branches, and their numbers scale with overall animal and brain size. Because these cells adopt a planar and well-spread configuration, quantification of whole animal *cto* cell number through automated image segmentation is straightforward and robust (*Oviedo et al., 2003*; *Hill and Petersen, 2015*). Decapitated animals which initially lack *cto+* cells regenerate them as they form a new head until an appropriate ratio of *cto+* cells to body size is attained. We treated animals with dsRNA and amputated to produce head, tail, and trunk fragments that were fixed and stained in 96-well mesh plates after regeneration. In order to maximize the effects of RNAi, regenerating trunk fragments were amputated again and scored for their ability to regenerate *cto+* cells after the second round of regeneration, whereas tail and head fragments were only fixed and scored after a single round of regeneration (*Reddien et al., 2005*). After imaging, numbers of *cto+* cells and total animal area were measured using a CellProfiler pipeline, and differences in relative *cto+* cell regeneration due to RNAi treatment were determined by

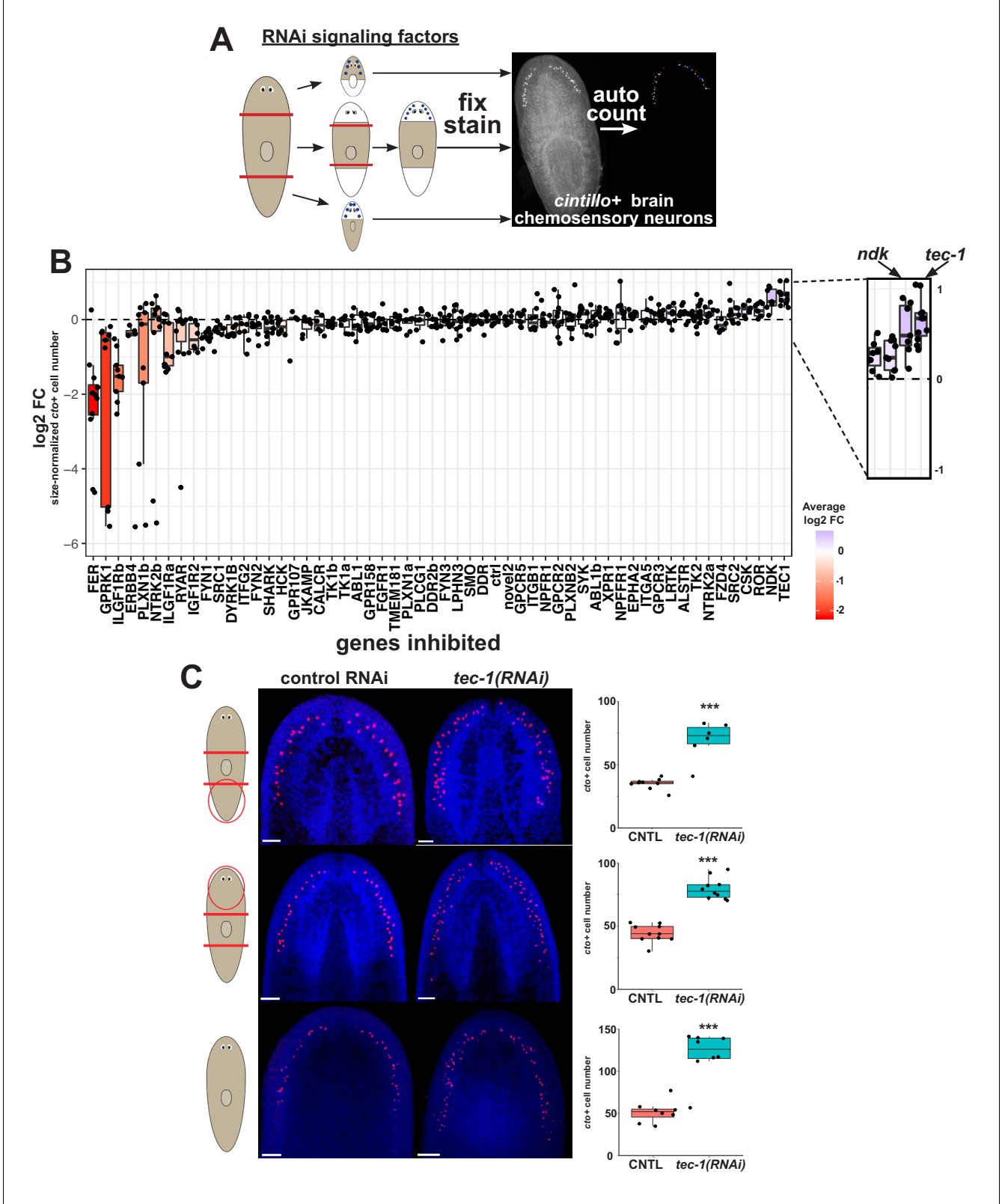

**Figure 1.** An RNAi screen identifies regulators of neuronal regeneration. (**A**) Design of RNAi screen. Animals were fed over several weeks, challenged to regenerate after head and tail amputation, and then fixed and stained for *cto* expression using Hoechst as a counterstain to detect total animal tissue (heads and tail fragments fixed at d23, while and regenerating trunks were fed dsRNA at d10 then amputated again the following day and then fixed 12 days later). Numbers of *cto+* cells were enumerated using CellProfiler and normalized to animal size as measured by Hoechst area in

*Figure 1 continued*

CellProfiler. (B) Log2-transformed fold changes of area-normalized *cto*-cell number were determined by comparison to similar fragments treated with control dsRNA, combined across head, trunk and tail fragment types for each dsRNA treatment and plotted in ascending order of average log2-fold change. Dots represent log2-fold change for each regenerating fragment, with boxplot shading representing average log2-fold change compared to controls. Knockdown of *tec-1* increased *cto+* cell number more than *nou-darake* positive control on average. Dotted line indicates log2FC for control RNAi conditions. (C) To measure *tec-1* RNAi's effect on head regeneration and injury-induced remodeling, animals were fed dsRNA for 2 weeks before decapitation and allowed to regenerate for 16 days. To measure the effects of *tec-1* inhibition in homeostasis, animals were fed dsRNA for 60 days and fixed without injury. An increase in *cto+* cell number was detected in all contexts. ***p<0.001 by two-tailed t-test. Scale bars: 100 µm.

The online version of this article includes the following figure supplement(s) for figure 1:

**Figure supplement 1.** *tec-1* mRNA is depleted by dsRNA feeding.
**Figure supplement 2.** *tec-1* and *tec-2* encode Tec family kinases.
**Figure supplement 3.** *tec-1* RNAi does not alter body size.
**Figure supplement 4.** *tec-2* does not act with *tec-1*.

computing a normalized log2-fold change ratio as compared to groups treated with a non-targeting dsRNA (*C. elegans unc-22*). Treatment with a positive control dsRNA targeting *ndk*, a factor that restricts head regionalization in planarians (*Cebrià et al., 2002*), produced increased numbers of *cto* + cells in the screen. The screen identified four genes whose inhibition reduced numbers of *cto+* cells attained through regeneration (log2FC < −1 and padj <0.05), though several other treatments caused smaller *cto+* cell decreases that did not meet the false-discovery corrected statistical cut-off (*Supplementary file 2*). Because these factors could either be required for expression of *cto* in mature neurons, be required for differentiation from neural progenitors, or be required for maintenance of neoblasts in general, we did not pursue any further analysis of them here (*Figure 1B*).

We sought to uncover factors that negatively regulated *cto+* cell production and found one factor, a Tec-family kinase (*tec-1*), whose inhibition caused a statistically significant increase in abundance of *cto+* cells (*Figure 1B*, *Figure 1—figure supplement 1*, *Figure 1—figure supplement 2*, *Supplementary file 2*). To verify these effects, we inhibited *tec-1* in a similar design as the screen but with an increased sample size. These experiments confirmed that inhibition of the Tec homolog resulted in a robust ~50–100% increase to the number of *cto+* cell numbers, which had approximately normal spatial distribution in the animal (*Figure 1C*). *tec-1* knockdown led to an increase in *cto+* cell number both in decapitated animals regenerating an entirely new head and also in head fragments undergoing tissue remodeling (*Figure 1C*). This increase in relative *cto+* cell abundance was not due to a change in overall animal size, but rather to an increase in absolute numbers of *cto+* cells (*Figure 1—figure supplement 3*). Regeneration of head and tail fragments involves extensive stem cell-dependent production of new brain neurons and control of the rates of cell death, as well as injury-induced signals that initiate the process. Uninjured planarians undergo perpetual homeostatic regeneration of all cell types, including neurons of the brain, but lack expression of injury induced factors. In order to test whether *tec-1*'s function depended on wound signaling, we fed animals *tec-1* dsRNA for 60 days in the absence of injury and found that *cto+* cell number increased to a similar extent as in amputated fragments (*Figure 1C*). Therefore, *tec-1* functions to limit numbers of neurons independent of injury signaling.

To examine a possible redundancy of function, we scanned the *Schmidtea mediterranea* genome and transcriptome and identified a second Tec family kinase, which we named *tec-2* (*Figure 1—figure supplement 2*). Inhibition of *tec-2* by RNAi did not increase *cto+* cell numbers or enhance the effects of *tec-1* RNAi (*Figure 1—figure supplement 4*). Therefore, *tec-2* likely does not act redundantly with *tec-1* for control of *cto+* cell abundance.

RNAi phenotypes that modify body patterning (such as *nou darake* RNAi) can result in production of excess *cto+* cells as a consequence of axis transformation (*Gurley et al., 2008*; *Iglesias et al., 2008*; *Petersen and Reddien, 2008*; *Lander and Petersen, 2016*; *Scimone et al., 2016*), so we tested whether *tec-1* could operate similarly. Whereas *ndk* inhibition increased *cto+* cell number concomitant with increasing the size of the brain (*Cebrià et al., 2002*), *tec-1* inhibition did not alter brain size (*Figure 2A*). Instead, *tec-1* RNAi, but not *ndk* RNAi, significantly increased the number of *cto+* cells normalized to brain size. The observation that *tec-1* inhibition increased numbers of *cto+* neurons without altering the relative size of the head or brain raised the question of how such

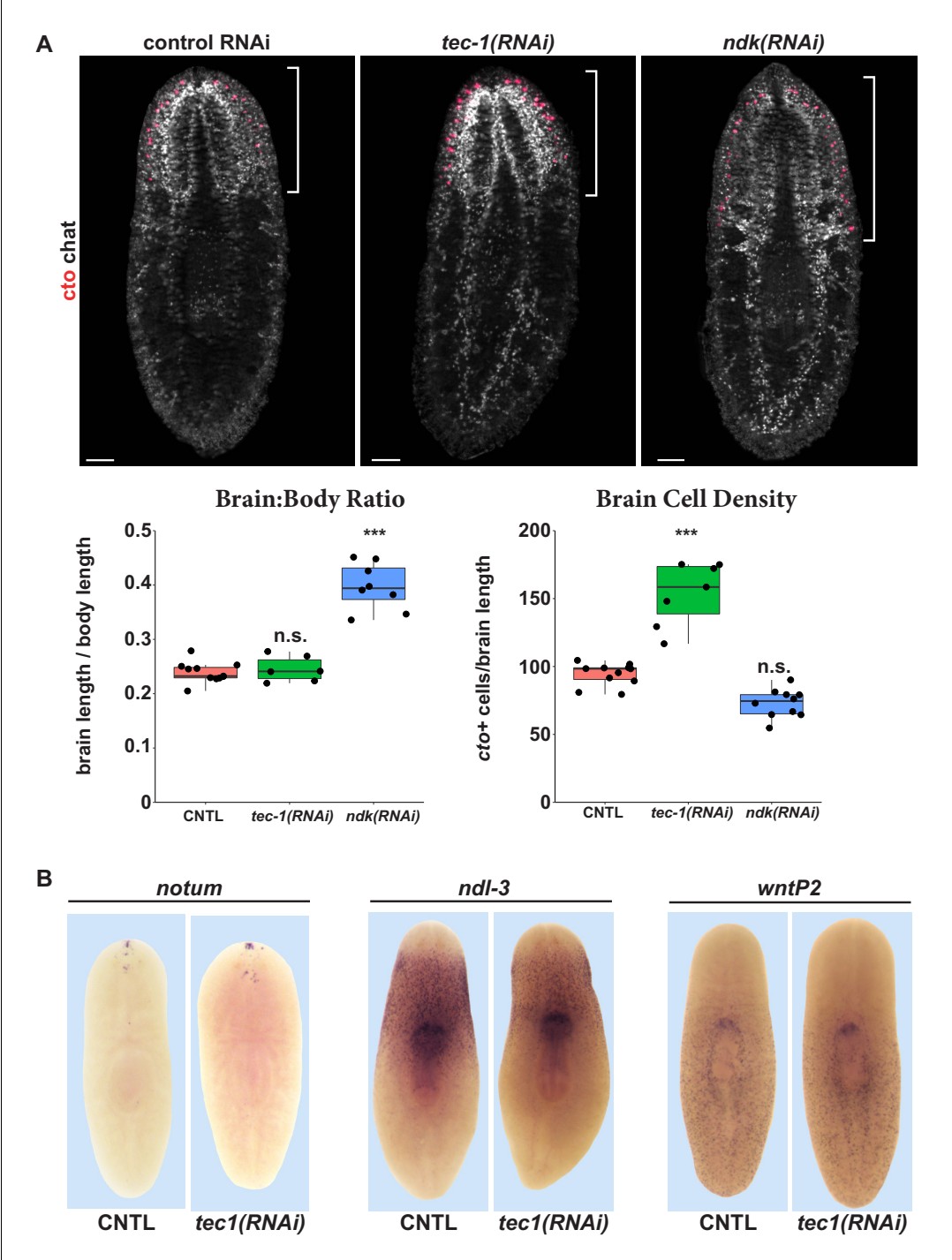

**Figure 2.** *tec-1* inhibition increases chemosensory neuron abundance but not head regionalization. (**A**) Animals were fed dsRNA for 2 weeks, amputated to remove heads and tails, and allowed to regenerate for 16 days. FISH was performed on trunk fragments to simultaneously measure *cto+* chemosensory neurons and *chat+* cholinergic neurons that allow visualization of the brain. Effects on the brain:body size and density of *cto+* cells normalized to brain size were quantified. *ndk* RNAi resulted in an increase in brain:body size but not in numbers of *cto+* cells normalized to brain size. By contrast, *tec-1* RNAi increased density of *cto+* cells within the brain without increasing the proportion of the body axis occupied by the brain. (**B**) *tec-1* RNAi did not strongly affect the expression of anterior-posterior positional control genes *notum*, *ndl-3*, and *wntP-2* as measured by WISH (images representative for 27/27 animals probed). Scale bars: 100 μm.

The online version of this article includes the following figure supplement(s) for figure 2:

**Figure supplement 1.** *tec-1* inhibition reduces the nearest-neighbor distance and volume of *cto+* neurons.

*Figure 2 continued on next page*

*Figure 2 continued*

**Figure supplement 2.** Excess *notum*+ neurons in the brain after *tec-1* inhibition.

animals physically pack increased neurons. To clarify how planarian brains could increase the number of cells without altering their overall size, we mapped the 3D positions and sizes of *cto*+ cells in control versus *tec-1(RNAi)* animals, and determined their nearest-neighbor distances and cell body volumes using Fiji/ImageJ (*Figure 2—figure supplement 1*). *tec-1* inhibition decreased the average nearest-neighbor distance of *cto*+ cells and also the average volume of each cell as measured by *cto* mRNA distribution. Therefore, *tec-1* influences *cto*+ cell abundance and size.

To confirm a lack of overall changes to body regionalization in *tec-1(RNAi)* animals, we examined positional control genes whose expression demarcates distinct body territories. *ndl-3* and *wntP-2* expression was not appreciably altered in *tec-1* RNAi treatments that result in excess *cto*+ cells (*Figure 2B*). By WISH, *tec-1* inhibition was observed to modestly alter the expression of the anteriorly expressed gene *notum*, but through FISH analysis this effect could be attributed to the expansion of a population of previously identified *notum+chat+* neurons present in the anterior of the brain (*Figure 2B*, *Figure 2—figure supplement 2*). (*Hill and Petersen, 2015*). We conclude that *tec-1* limits numbers of *cto*+ cells by altering neuron density rather than participating strongly in body patterning, and that *tec-1* regulates the abundance of more than one type of neuron.

## *tec-1* negatively regulates the abundance of many neural cell types

In order to test the specificity *tec-1*'s function to limit differentiated cell abundance, we investigated the impact of *tec-1* knockdown on other neural cell types whose abundances could be quantified with high precision. We began by examining neurons expressed more medially within the brain compared to chemosensory neurons. *glutamic acid decarboxylase* (*gad*) is expressed in GABAergic neurons in the ventral-medial and dorsal-lateral CNS (*Nishimura et al., 2008b*). Neurons in these domains have distinct progenitor populations expressing *nkx2.1* and *tcf1* respectively (*Currie et al., 2016*; *Brown et al., 2018*). Inhibition of *tec-1* increased the total number of GABAergic cells and also numbers of cells within both domains (*Figure 3A* and *Figure 3—figure supplement 1*), suggesting that *tec-1*'s activity is not restricted to a single brain region or a specific neuronal lineage. To confirm this, we quantified cells found throughout the brain which expresses the neuropeptide precursor-encoding gene *pyrokinin prohormone-like 1* (*ppl-1*) (*Collins et al., 2010*). Cephalic *ppl-1*+ cells increased in abundance after *tec-1* knockdown (*Figure 3A*). *ppl-1* is also expressed prominently in neurons of the pharynx, but these cells were unaffected by *tec-1* RNAi (*Figure 3B*), indicating *tec-1* does not regulate abundance of all neurons throughout the animal. We additionally examined serotonergic neurons in the CNS marked by *serotonin transporter* (*sert*), which have a defined progenitor cell type (*Currie and Pearson, 2013*; *März et al., 2013*), and *dd17258*+ neurons expressed in a domain similar to *cto*+ cells (*Fincher et al., 2018*). We observed that the numbers of both of these populations increased upon *tec-1* inhibition. To test whether *tec-1* might act exclusively within the CNS, we quantified the density of nociceptory *trpA*+ neurons (*Wenemoser et al., 2012*; *Arenas et al., 2017*) and peripheral cholinergic *chat*+ neurons (*Nishimura et al., 2010*) of the in the PNS and found that *tec-1* RNAi increased the abundance of both cell types (*Figure 3A*). To obtain more information on the effects of *tec-1* knockdown throughout the body axis, we also sought neuron cell types distributed throughout the body and with densities and abundances amenable to whole-animal enumeration. We selected four such cell types that had been identified in a prior scRNAseq cell atlas study (*Figure 3—figure supplement 2*) (*Collins et al., 2010*; *Fincher et al., 2018*). *tec-1* knockdown increased the density of three of these cell types: *dd2223*+, *dd3733*+, and *spp-4*+ neurons. However, *tec-1* knockdown did not affect the abundance of *dd2723*+ neurons, which are expressed in the CNS and pharynx (*Figure 3—figure supplement 2A*). In addition, we also examined photoreceptor neuron abundance, because *tec-1(RNAi)* animals regenerating a new head sometimes produced disorganized eyes (*Figure 3B*). Despite affecting eye morphology, *tec-1* inhibition did not modify the number of photoreceptor neurons. Together, we found that *tec-1* inhibition increased the abundance of 10 of 13 neuron markers investigated, indicating that *tec-1* negatively regulates the abundance of many but not all types of neurons within the central and peripheral nervous systems.

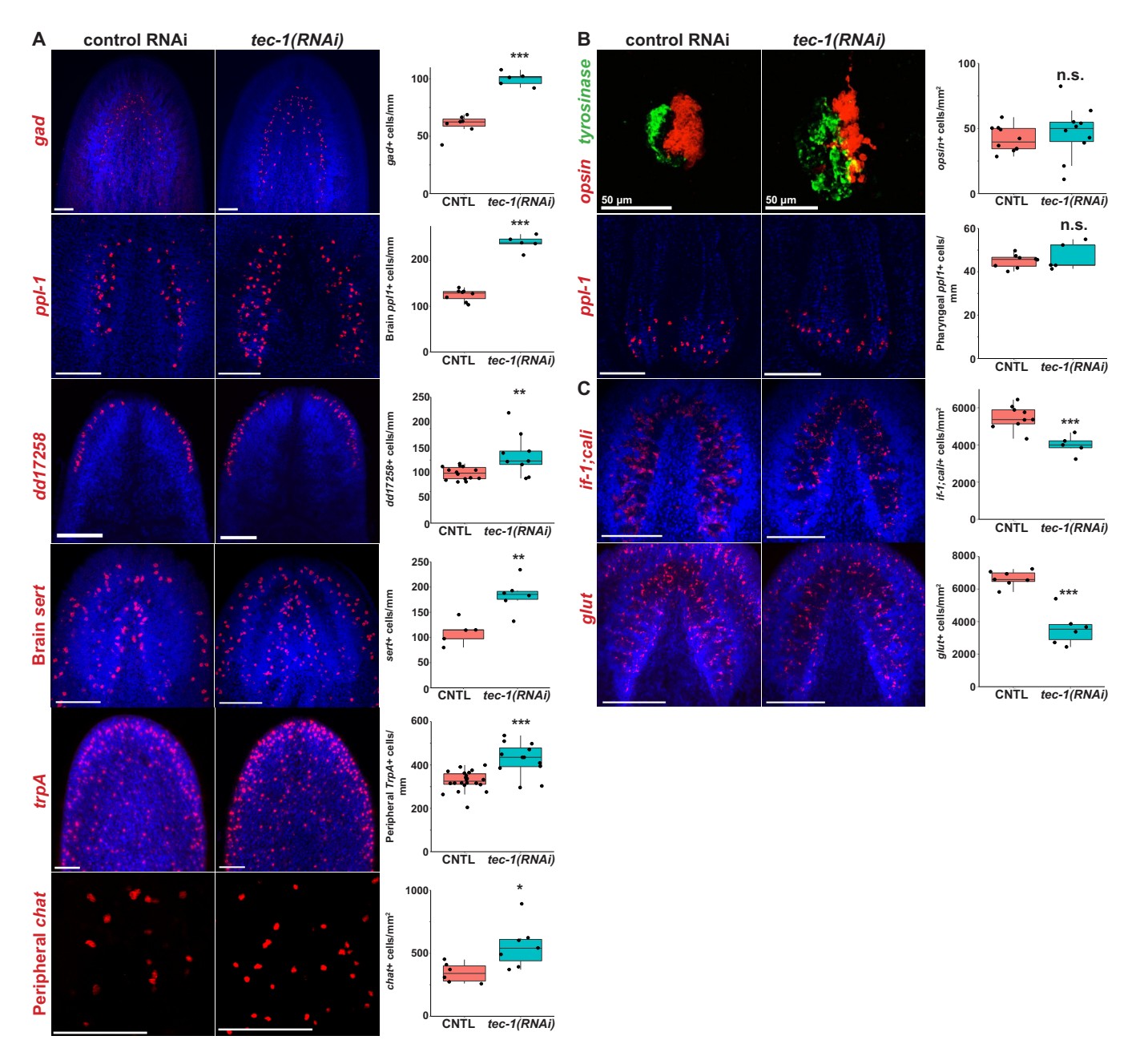

**Figure 3.** *tec-1* inhibition increases abundance of several CNS and PNS neurons. (**A-C**) Animals were fed dsRNA for two weeks, amputated to remove heads and tails, and allowed to regenerate for 15 days. Regenerating head fragments were fixed and stained for *gad* expression, regenerating tail fragments were stained for *ppl-1, trpA, opsin/tyrosinase, glut,* or *if-1/cali* expression, and regenerating trunks were stained for *chat* or *dd17258* expression. Cell types amenable to total animal enumeration were quantified by normalizing cell number to body size by dividing by the square root of whole animal area (*gad*+, brain and pharynx *ppl1*+, brain *sert*+, *dd17258*+, *opsin*+ cells). Abundances of cell types too numerous for whole-body counting were quantified by manually scoring cell numbers in a region of interest and normalizing to the area of that region (*trpA*+ cells were scored in an anterolateral region, peripheral *chat*+ neurons scored in a postpharyngeal ventromedial region, *if-1*+;*cali,* and *glut*+ cells scored within the brain defined by Hoechst staining). (**A**) *tec-1* RNAi animals had increased numbers of *gad*+ neurons, *ppl-1*+ neurons within the brain, *dd17258*+ neurons, *sert* + brain neurons, *trpA*+ peripheral neurons, and *chat*+ peripheral neurons. (**B**) *tec-1(RNAi)* animals regenerated disorganized photoreceptors but had no alteration in *opsin*+ cell numbers. Likewise, *tec-1* inhibition did not alter numbers of pharyngeal *ppl-1*+ neurons. (**C**) *tec-1* knockdown decreased the density of *glut*+ and pooled *if-1/cali*+ glial cells in the brain. Significance determined by two-tailed t-tests (*, p<0.05; ***, p<0.001; n.s. p>0.05). Images show Hoechst-stained maximum projections except for maximum-projected pharyngeal *ppl-1*+ cells shown for clarity overlayed with a single slice of Hoechst-labeled pharynx tissue. Scale bars: 100 μm unless otherwise noted.

*Figure 3 continued on next page*

*Figure 3 continued*

The online version of this article includes the following figure supplement(s) for figure 3:

**Figure supplement 1.** Two GABAergic neuron populations are regulated by *tec-1*.
**Figure supplement 2.** Effects of *tec-1* inhibition on neurons expressed across the A-P axis.
**Figure supplement 3.** Additional analysis of neuron density in *tec-1* RNAi.
**Figure supplement 4.** *Tec-1(RNAi)* neurons form projections.
**Figure supplement 5.** Non-neural cell types not affected by *tec-1* RNAi.

Because many of these effects were identified in brain-localized, anterior cell types, we wanted to understand whether the *tec-1* RNAi phenotype only affected anterior cells. Planarian proliferative cells have differential sensitivities to DNA damage repair and cell death across the A-P axis (*Peiris et al., 2016*). To test a potential influence of axis position on the *tec-1* RNAi phenotype, we determined the A-P position of all *dd2223*+ cells in control and *tec-1* RNAi animals and binned these into anterior and posterior regions (*Figure 3—figure supplement 2B*). *tec-1* inhibition increased the abundance of *dd2223*+ neurons in both the anterior and posterior of the animal. Furthermore, *tec-1* RNAi did not alter the relative proportion of anterior versus posterior *dd2223*+ neurons. Together, we conclude that *tec-1* can act equivalently on neurons independent of body position.

We also wanted to determine whether *tec-1* exerts its influence equally on distinct cell types. To test this, we stained *tec-1(RNAi)* and control animals simultaneously for *cto* and either *gad* or *ppl-1*, which are expressed in different areas of the brain. Counting two neuron populations at once allowed direct comparisons of the *tec-1* RNAi phenotype's expressivity for each animal. *tec-1* inhibition increased numbers of these neuron cell types to the same extent without affecting *gad:cto* or *ppl1:cto* cell ratios (*Figure 3—figure supplement 3*). Therefore, *tec-1* appears to affect multiple neuron type abundances concordantly.

We next sought to determine whether excess neurons produced in *tec-1(RNAi)* animals had features of normal neurons. We noticed that riboprobes detecting *cto* mRNA primarily stained the cell body, but high magnification imaging revealed some detection of signal in processes extending laterally from these bodies, putatively related to axonal or dendritic regions. A similar proportion of *cto*+ cells had such processes in control versus *tec-1(RNAi)* animals (*Figure 3—figure supplement 4A*). We also used alpha-tubulin staining to examine overall innervation patterns. In this assay, *tec-1 (RNAi)* animals displayed a qualitative increase in size of nerve bundles, particularly evident in the transverse commissural fibers joining the nerve cords along the ventral side of the animal (*Figure 3— figure supplement 4B*, 6/7 animals examined). Treatments disrupting synaptic transmission can cause failure of neuron regeneration in planarians (*Inoue et al., 2007*), suggesting that the excess neurons in such animals may be functional.

We reasoned that *tec-1* might control cell abundance specifically within a subset of CNS and PNS neurons or more generally throughout the animal. Neither *collagen*+ body wall muscle cells nor epidermal nuclei were found to have statistically significant changes in density after *tec-1* inhibition (*Figure 3—figure supplement 5A*), suggesting the specificity of *tec-1*'s cell number control activity to the nervous system. We also examined animals stained for broad markers of excretory and intestinal tissue (*Figure 3—figure supplement 5B*). There were not obvious increases in expression of either marker of these tissue types.

In addition, we also examined non-neuronal cells in the CNS characterized as astrocyte-like glial cells (*Roberts-Galbraith et al., 2016*; *Wang et al., 2016*). *tec-1* inhibition decreased the density of cells expressing pooled glial markers *intermediate filament 1* (*if-1*) and *calamari* (*cali*) as well as those expressing *glucose transporter* (*glut*) (*Figure 3C*). The functions for planarian glial cells are not yet characterized, but these cells could have a role in neuron surveillance.

## *tec-1* suppresses neuronal cell number by regulating cell survival

We next sought to determine how *tec-1* exerts its negative regulatory function on neuron abundance. Differentiated cells are produced continually from neoblasts in adult planarians, so that increases to numbers of differentiated cells could either arise from an increase in stem cell-dependent tissue production or through decreases in the death of differentiated cells. Because *tec-1* knockdown broadly increased neuronal numbers while decreasing glial numbers, and it is known

that neurons and glial cells can arise from common progenitor cells in some contexts in other organisms, we initially hypothesized that *tec-1* might control a differentiation switch at some early point in neural specification (*Mori et al., 2005*; *Homem and Knoblich, 2012*).

To better understand the dynamics of *tec-1*'s effects on regeneration, we measured *cto+* cell abundance over time during head regeneration after decapitation and during remodeling of head fragments (*Figure 4A*). In decapitated *tec-1(RNAi)* animals regenerating a new head, the rate of production of new *cto+* cells was not altered at early times (4 days), but abundance of *cto+* cells was higher after a week of regeneration (8 days) and reached a maximum point at 16 days. We also confirmed through WISH that *tec-1* is expressed broadly in the animal and throughout regeneration from day 2 to day 12 post-amputation without undergoing discernable regeneration-induced expression changes (*Figure 4—figure supplement 1*). Given this expression and that *tec-1* affects neuron density in both regeneration and homeostasis, it is likely this gene functions constitutively to regulate neuron numbers. To test for the perdurance of the phenotype in regeneration, we measured *cto+* cell number up to 4 weeks after decapitation and found that *cto* cells had reached a steady-state maximum by two weeks post-amputation (*Figure 4—figure supplement 2*). In *tec-1 (RNAi)* head fragments undergoing tissue remodeling, the maximal *cto+* cell abundance phenotype was observed somewhat earlier, by 4 days, and persisted (*Figure 4A*). The progressive nature of the phenotype in decapitated regenerating animals suggests that *tec-1* is unlikely to regulate rates of differentiation commonly across all timescales and conditions. However, it remained possible that rates of neuron differentiation are maximal during early head regeneration, while *tec-1* could restrict rates of differentiation in a process common to the contexts of late head regeneration, head remodeling, and homeostatic maintenance.

To test this possibility, we designed a BrdU labeling strategy to test for possible effects of *tec-1* inhibition on the rate of neuron differentiation. We chose to investigate regenerating head fragments undergoing brain remodeling because *tec-1* inhibition caused a rapid attainment of excess neurons compared to control animals, so we reasoned these fragments would provide a context in which *tec-1*-dependent regulation is prominent. Neoblasts are the only proliferative cells in adult planarians, so a pulse of BrdU initially marks these cells, followed by labeling their newly-born postmitotic descendants (*Newmark and Sánchez Alvarado, 2000*). In order to measure rates of neuron production, we soaked animals with BrdU earlier on same the day of amputation (d0) or at 2 or 4 days after amputations to generate head fragments, then fixed the animals at 6, 8, 10 and 12 days post-amputation in order to detect BrdU incorporation into new mature neurons (*Figure 4B*). We chose to analyze BrdU incorporation into *ppl-1+* brain neurons because they are under robust control by *tec-1* (*Figure 3A*), and they are more numerous than cell types such as *cto+* or *gad+* neurons, thus maximizing the ability to detect any influence of *tec-1* on neuron differentiation given possible inefficiencies in BrdU label uptake. However, numbers of *ppl-1+*BrdU+ cells were not significantly different in control versus *tec-1(RNAi)* animals over a range of timepoints of BrdU pulsing and fixation in regeneration. These data indicate that *tec-1* inhibition can increase numbers of neurons without modifying rates of differentiation, pointing instead to a function in controlling neuron survival.

A model in which *tec-1* controls neuronal survival would predict that *tec-1* does not influence the abundance of neural progenitor cells that are the intermediates between pluripotent neoblast stem cells and differentiated neurons (*Figure 4C*). *coe* encodes a transcription factor expressed in a subpopulation of *piwi-1+* neoblasts as well as in a variety of differentiated neurons, and its inhibition reduces the abundance of several types of neurons, including *gad+* and *cto+* neurons (*Cowles et al., 2013*; *Cowles et al., 2014*). *tec-1* inhibition did not alter numbers of *coe+piwi-1+* cells in a timeseries between 2 and 12 days of head regeneration, despite detection of an overall decreasing abundance of such progenitors in both control and *tec-1(RNAi)* animals over time as regeneration proceeds. To confirm these observations, we examined a subpopulation of *piwi-1+* neooblasts expressing the proneural transcription factor *pax6* (*Scimone et al., 2014*) and found that its abundance was also unaltered by *tec-1* inhibition in the same time period. *coe* and *pax6* mark broad classes of neural progenitors, so we also sought to test the abundance of neural progenitors that produce individual neural cell types. *pitx* transcription factor is required for production of *sert+* serotonergic neurons and *pitx* is expressed within a *piwi-1+* neoblast subpopulation proposed to be the progenitors for these cells. Although *tec-1* inhibition increased the abundance of *sert+* neurons in the brain (*Figure 3A*), *tec-1* RNAi did not alter numbers of *pitx+piwi-1+* serotonergic neuron progenitors near the brain (*Figure 4C*).

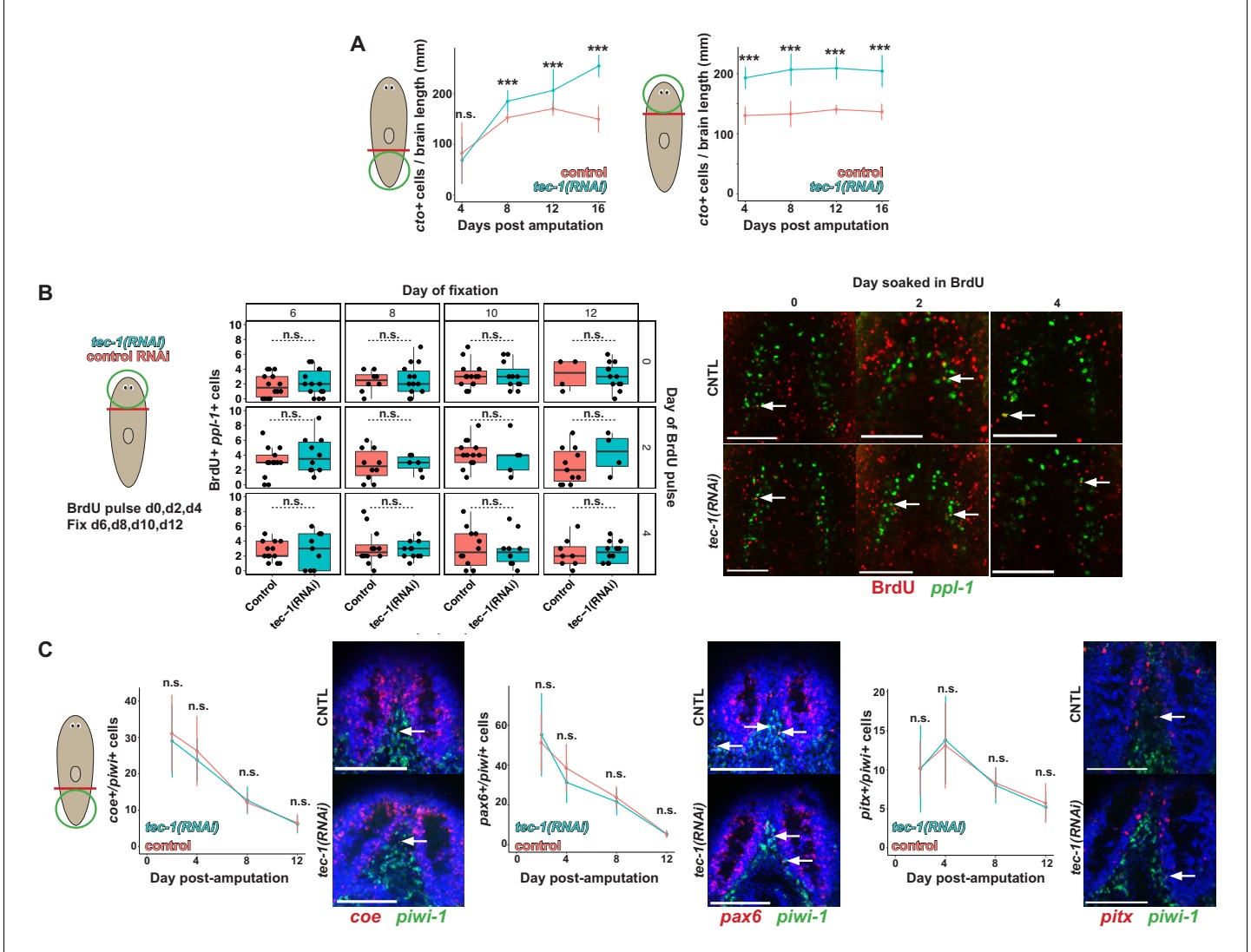

**Figure 4.** *tec-1* inhibition does not increase rates of neuron differentiation. All animals were fed dsRNA for 2 weeks before surgeries as indicated by cartoons. (A) Time courses of *cto+* cell number in regeneration of a new head (left) or remodeling of a pre-existing head (right). Each data point represents sample size of between 4 and 12 animals. (B) Animals were transtioned into high-salt water one week before surgery, and either whole animals (day 0) or head fragments (days 2 and 4) were soaked in BrdU for 4 hr. Heads fragments were fixed at days 6, 8, 10, and 12 post-amputation. Co-staining of control versus *tec-1(RNAi)* animals for *ppl-1* mRNA and BrdU showed no detectable increase in numbers of BrdU+*ppl-1*+ cells at any time point (left). Single confocal slices show staining of BrdU and *ppl-1* in day 12 animals (right). White arrows show colocalization. (C) Regenerating tail fragments were fixed at 2, 4, 8, or 12 days post injury and stained with FISH to detect *piwi-1* and *coe* or *pax6* transcription factors, which label broad neuronal progenitors, or *pitx* transcription factor, labeling progenitors of serotonergic neurons. Single confocal slices show staining of transcription factors, *piwi-1*, and Hoechst counterstain in day 12 animals. No significant differences in numbers of neural progenitor cells were detected (***, p<0.001, n.s. p>0.05 by two-tailed t-test, error bars represent standard deviation). Scale bars: 100 μm.

The online version of this article includes the following figure supplement(s) for figure 4:

**Figure supplement 1.** *tec-1* is expressed during regeneration.
**Figure supplement 2.** *tec-1* RNAi phenotype reaches a steady state by 2 weeks of head regeneration.
**Figure supplement 3.** *tec-1* is broadly expressed in neural cells and progenitors.
**Figure supplement 4.** *tec-1* RNAi does not affect mitotic activity 48 hr after injury.

We next examined *tec-1* expression by double-FISH to determine whether this gene could be expressed preferentially within neurons, neural progenitors, or other cell types. However, we detected *tec-1* FISH signal not only within each type of neuron tested (*cto*, *gad*, *ppl-1*) but also within *pitx*, *coe*, or *pax6A*-expressing cells, neoblasts or their differentiating progeny marked with

anti-PIWI-1 antibody, and in *if-1;cali*+ glial cells (*Figure 4—figure supplement 3*). Therefore, *tec-1* is a broadly expressed gene that exerts a specific effect on neuronal cell density. Consistent with these observations and the specificity of the *tec-1(RNAi)* phenotype, *tec-1* inhibition did not modify numbers of H3P+ mitotically active cells in response to injury (*Figure 4—figure supplement 4*). Taken together, these results support the model that *tec-1* does not regulate the process of neuronal differentiation or control a switch between neuronal and glial specification.

In light of these results, we reasoned that *tec-1* might function to activate cell death specifically of CNS and PNS neurons. Uninjured planarians undergo a basal rate of homeostatic cell death thought to occur across tissue types as older cells die off and require stem cell-dependent replacement, and amputation triggers a systemic elevation of cell death of differentiated cells as part of the tissue remodeling process (*Pellettieri et al., 2010*). We used TUNEL staining on remodeling head fragments to test whether *tec-1* promotes cell death and found that *tec-1* RNAi reduced but did not eliminate numbers of TUNEL+ cells (*Figure 5A*). This observation is consistent with a model in which *tec-1* promotes cell death of some but not all cells.

We hypothesized based on the above results that *tec-1* might promote cell death in a fraction of newly born neurons. The requirement of *tec-1* under conditions of both homeostasis and regeneration would then suggest that, as in embryonic development in other organisms, regenerated brain neurons in planarians undergo an initial overproduction followed by rapid cell death. The regeneration of a new head in decapitated animals represented a condition in which we reasoned this phenomenon would be most easily detected, because in that context the new production of brain neurons is synchronized by amputation. However, detailed time course analysis of new neuron production in such animals has not yet identified such overproduction, suggesting that ongoing neuron production during the ~1–2 weeks of head regeneration might mask these effects. Therefore, we sought a means to isolate the fates of a limited cohort of neurons produced only within a narrow time window after amputation. To do this, we used lethal irradiation early during head regeneration to eliminate neoblasts acutely, followed by a timeseries of fixation and staining, to measure the survival of a cohort of neurons produced only within in a specific time frame. After *tec-1* inhibition and head amputation, planarian fragments were irradiated 6 days later then fixed in a time series in the absence of proliferating cells. Staining these animals with a *piwi-1* riboprobe 2 days after irradiation showed complete elimination of neoblasts by this time (*Figure 5—figure supplement 1*). In control animals, *cto*+ cell number decreased between days 10 and 12 post-amputation, consistent with the prediction of initial overproduction (*Figure 5B* and *Figure 5—figure supplement 2*). *tec-1(RNAi)* animals did not produce significantly more *cto*+ cells at day 10 compared to controls but did not undergo the day 12 decrease in *cto*+ cells. These data confirm that *tec-1* does not influence rates of cell production but instead strongly suggest that *tec-1* promotes the death of newly born neurons in order to limit the abundance of cells produced through adult neurogenesis (*Figure 5C*).

## Discussion

Here we identify the conserved gene *tec-1* as a potent and novel negative regulator of stem cell-dependent adult neurogenesis in *Schmidtea mediterranea*. Completion of regeneration and homeostatic maintenance requires control of both the rates of new cell differentiation and also control of cell survival (*LoCascio et al., 2017*). Several positively acting factors contribute to differentiation of new neurons in planarians. Injuries trigger expression of the *runt-1* transcription factor within neoblasts, required for producing normal numbers of *trpA*+ brain cells, eyes, and other cells. Other fate determinants appear to act constitutively to control differentiation or expression within distinct neuronal populations (*Wenemoser et al., 2012*; *Cowles et al., 2013*; *Currie and Pearson, 2013*; *März et al., 2013*; *Scimone et al., 2014*; *van Wolfswinkel et al., 2014*; *Currie et al., 2016*; *Molinaro and Pearson, 2016*; *Brown et al., 2018*; *Fincher et al., 2018*).

By contrast, fewer regulators of cell death have been identified in planarians and these appear to regulate multiple lineages. Inhibition of *bcl2-1*, a conserved antiapoptotic factor, causes a rapid increase in systemic cell death and subsequent animal lysis (*Pellettieri et al., 2010*). *jnk* and *yorkie* inhibition both decrease cell death and increase numbers of *cto*+ cells relative to overall body size in decapitated head fragments (*Almuedo-Castillo et al., 2014*; *Lin and Pearson, 2014*). However, in both of these phenotypes, an increase in neuronal number is linked to an increase in overall brain size and is in a context in which the brain normally undergoes reduction through injury-induced

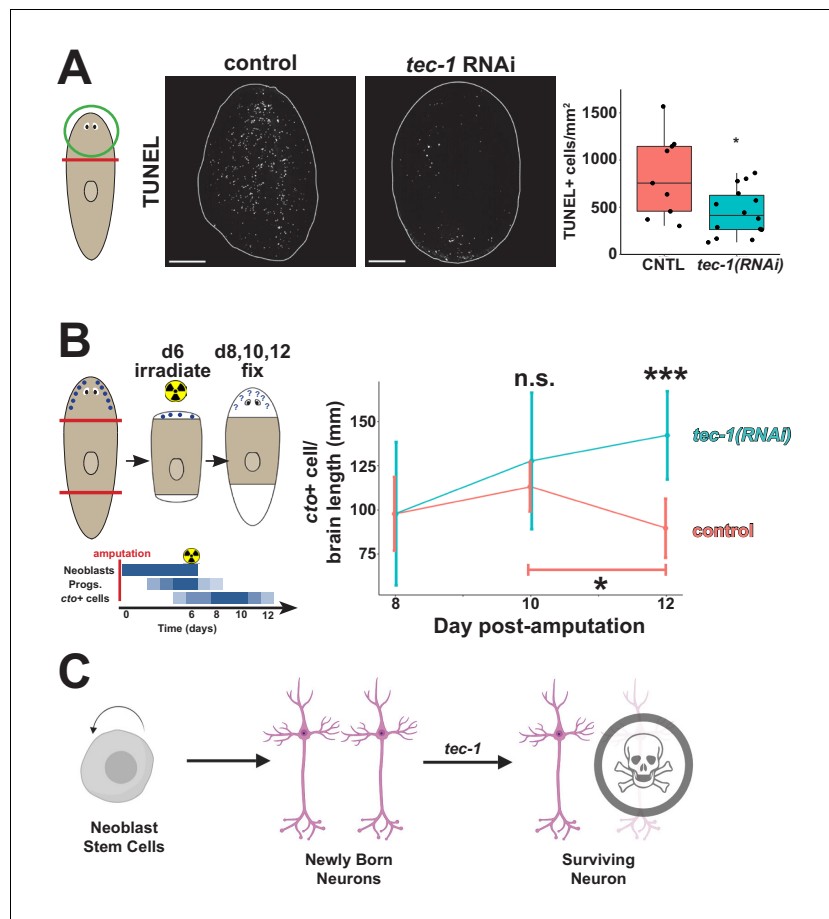

**Figure 5.** *tec-1* promotes cell death and limits survival of new neurons. Animals were fed dsRNA for 2 weeks before amputation to measure numbers of dying cells and persistence of regenerated chemosensory neurons after lethal irradiation. (**A**) Head fragments were TUNEL stained 72 hr after injury and numbers of TUNEL+ cells normalized to fragment area quantified to find that *tec-1* inhibition resulted in diminished numbers of dying cells. (**B**) Amputated trunk fragments were allowed to regenerate for 6 days, treated with 6000 rads of X-rays to eliminate neoblasts and subsequent neuron differentiation, then fixed at the indicated times post-injury, and numbers of *cto+* cells detected and counted by FISH and normalized to brain length determined by Hoechst staining. Schematic shows predicted effects of lethal irradiation on population abundances (blue shading) for cells involved in producing new *cto+* neurons in head regeneration: neoblasts, neural progenitors, and differentiated CNS neurons after irradiation during head regeneration over the course of days. In control animals, the density of *cto+* neurons (red) decreased between four and six days after irradiation (bracket). By contrast, *tec-1(RNAi)* animals had normal numbers of newly formed *cintillo+* cells, *cto+* number is not significantly greater than controls at d10 post amputation, but is significantly greater than controls two days later. Each data point represents sample size of between 4 and 9 animals. (**C**) Model of normal neuronal production, where *tec-1* acts to cull excess neurons in homeostasis and regeneration (*p<0.05, **p<0.01, ***p<0.001, n.s. p>0.05 by two-tailed t-test, error bars represent standard deviation). Scale bars: 100 μm.

The online version of this article includes the following figure supplement(s) for figure 5:

**Figure supplement 1.** X-ray irradiation ablates neoblasts in *tec-1(RNAi)* animals.

**Figure supplement 2.** *cto* expression after irradiation in regenerating animals.

**Figure supplement 3.** *tec-1* is broadly expressed.

---

remodeling. Based on *tec-1's* specificity for neural cells and utilization in both regeneration and homeostasis, we suggest it operates in a distinct pathway (*Figure 5C*).

We envision at least two possible mechanisms by which *tec-1* facilitates neuronal cell death based on differing sites of action. *tec-1* could act cell autonomously within neurons to promote cell death or alternatively could be necessary within a distinct cell population involved in neuron killing and/or

engulfment. The broad expression of *tec-1* throughout multiple tissues would be consistent with either possibility (*Figure 4—figure supplement 3*). Cell atlas projects also found that *tec-1* is expressed broadly but is moderately enriched in neural progenitors, consistent with the first possibility, but also within intestine and *cathepsin+* cells that could have uncharacterized roles in neuron homeostasis (*Figure 5—figure supplement 3*) (*Fincher et al., 2018*; *Plass et al., 2018*). Future work delineating *tec-1* signaling targets could clarify this aspect of *tec-1's* anti-survival function.

Tec family kinases (TFKs) are non-receptor tyrosine kinases similar to Src family kinases. TFKs are perhaps best known for signaling downstream of antigen receptors in hematopoietic differentiation and lymphomas (*Berg et al., 2005*; *Rickert, 2013*). However, they are also involved in a wide variety of signaling pathways downstream of GPCRs, integrins, and both receptor and non-receptor tyrosine kinases, and are known to regulate apoptosis, cell adhesion, and the actin cytoskeleton in addition to differentiation (*Takesono et al., 2002*). In mammals, a majority of studies have found that Tec kinases such as BTK are essential for the survival or proliferation of immune cells, including phagocytic cells (*Takesono et al., 2004*; *Jongstra-Bilen et al., 2008*; *Melcher et al., 2008*; *Palmer et al., 2008*), suggesting that planarian *tec-1* might not act cell autonomously within neurons to promote death. Alternatively, in some contexts, Tec kinases have been reported to function as cell autonomous tumor suppressor genes that promote p53-mediated cell death pathways (*Rada et al., 2018*). Within the mammalian brain, chemical inhibition of the Tec-family kinase BTK increased brain cell survival after ischemic brain injury, an effect attributed to the modification of macrophage activity (*Ito et al., 2015*). It is possible that Tec kinases could regulate conserved functions in adult neurogenesis to control of neuron abundance in other organisms.

The requirement for *tec-1* for neuron suppression in regeneration and homeostasis in planarians additionally suggests that developmental cell death of neurons might be a common feature of adult neurogenesis across species. Cell death is extensively used throughout the animal kingdom to sculpt the developing nervous system, from nematodes to mammals (*Yamaguchi and Miura, 2015*). Among well-studied invertebrates, adult neurogenesis has been detected in *Drosophila* within the medulla cortex, where it can also mediate injury repair (*Fernández-Hernández et al., 2013*). However, the bulk of neurogenesis in *C. elegans* and *Drosophila* occurs at embryonic or larval stages prior to adulthood, so the precise roles of cell death for adult neurogenesis and ways they may be conserved across distant phyla have yet to be fully elucidated. Based on TUNEL staining and the observation that mouse mutants defective for cell death in the nervous system overproduce neurons in multiple regions of the brain, developmental cell death of neurons is integral to adult neurogenesis in mammals (*Biebl et al., 2000*; *Winner et al., 2002*; *Sun et al., 2004*), and cell death after differentiation has been confirmed through direct observation (*Pilz et al., 2018*). Regulation of cell death at the level of neural progenitors may control brain morphology by preventing hyperplasia or may define where crucial morphogenic signals are expressed (*Haydar et al., 1999*; *Nonomura et al., 2013*; *Yamaguchi and Miura, 2015*), while regulation of the survival of newly formed neurons has been proposed to provide mechanisms to ensure proper connectivity or limit interneuron cell number (*Southwell et al., 2012*; *Dekkers et al., 2013*). During brain development, preventing neurotransmitter secretion results in assembly of a mostly normal brain followed by massive neuronal apoptosis, suggesting that regulated cell death plays a role in the maturation of newly-differentiated neurons during embryogenesis (*Verhage et al., 2000*). Reliance on neurotransmitter signaling to promote survival of immature neurons is also used extensively in murine adult neurogenesis (*Petreanu and Alvarez-Buylla, 2002*), among other cell-type specific mechanisms (*Pfisterer and Khodosevich, 2017*). Many of these pathways converge on promoting survival through BCL2 family members, while death is promoted by Bax and Bak (*Motoyama et al., 1995*; *Shindler et al., 1997*; *Sun et al., 2004*; *Savitt et al., 2005*; *Nakamura et al., 2016*). Therefore, inhibitors of core cell death pathways have been proposed as therapeutic agents to assist in damage repair after brain injury (*Degterev et al., 2005*; *Loane and Faden, 2010*), though these would be expected to have substantial effects on other organ systems. If the ability of Tec kinases to limit neuron production ultimately proves to be conserved, they would be potential targets for enhancing neural repair in other species.

The identification of negative regulators of neurogenesis may be an important step in understanding neural regeneration in planarians and other systems. Suppressors of mTOR and JAK/STAT signaling in vertebrates can be inhibited to increase axon growth after spinal cord injury in laboratory conditions (*Park et al., 2008*; *Liu et al., 2010*; *Sun et al., 2011*). The JAK/STAT inhibitors Socs3

and Sfpq attenuate axonal regrowth in the optic nerve of mice and zebrafish (*Smith et al., 2009*; *Sun et al., 2011*; *Elsaeidi et al., 2014*). Additionally, while canonical Wnt signaling appears to promote neural regeneration in vertebrates (*Yin et al., 2008*; *Strand et al., 2016*; *Patel et al., 2017*), non-canonical Wnt signaling inhibits axon guidance and growth (*Onishi et al., 2014*). A more complete understanding of negative inputs into adult tissue homeostasis could provide new and more specific targets for the enhancement of neural tissue repair or treatment of degenerative disease (*Trounson and McDonald, 2015*). Our results suggest that planarians, which undergo extensive homeostasis of the CNS in adulthood, can be a model for efficiently identifying such negative regulators of adult neurogenesis.

# Materials and methods

## Key resources table

| Reagent type (species) or resource | Designation | Source or reference | Identifiers | Additional information |
|---|---|---|---|---|
| Gene (*Schmidtea mediterranea*) | Tec-1 | Planmine | dd_Smed_v6_4818_0_1 | |
| Antibody | Polyclonal rabbit anti-digoxigenin-POD, Fab fragments | Sigma/Roche | # 11207733910 RRID: AB_514500 | 1:2000 dilution |
| Antibody | Polyclonal rabbit anti-fluorescein-POD, Fab fragments | Sigma/Roche | #11426346910 RRID: AB_840257 | 1:2000 dilution |
| Antibody | Polyclonal rabbit digoxigenin-AP | Sigma/Roche | #11093274910 | 1:4000 dilution |
| Antibody | Rat polyclonal Anti-BrdU | Abcam | 6326 | 1:1000 dilution |
| Antibody | Rabbit monoclonal anti-phospho-ser10 Histone H3 | Cell Signaling | D2C8 | 1:3000 dilution |
| Antibody | Mouse monoclonal anti-TUBULIN-ALPHA AB-2 | Thermo/Fisher | MS581P1 | 1:1000 dilution |
| Commercial assay or kit | TUNEL labeling kit | Thermo/Fisher | EP0162 | N/A |
| Antibody | Goat polyclonal anti-rat HRP | Jackson ImmunoResearch | 112-036-072 | 1:1000 dilution |

## Planarian culture

Asexual strain CIW4 of the planarian *Schmidtea mediterranea* were maintained in 1 × Montjuic salts at 19°C as described (*Petersen and Reddien, 2011*). Animals were fed a liver paste and starved for at least 7 days before experiments.

## Screen

60 putative regulatory molecules were selected by analysis of the planarian transcriptome to identify a set with expression detected in neoblasts after FACS sorting (*Labbé et al., 2012*), (into populations of X1 neoblasts in G2/S/M phases, X2 cells comprised of a mixture of G1 neoblasts and G0 newly postmitotic progenitor immediate descendants of neoblasts or other irradiation-sensitive G1/G0 cells, and Xins differentiated cells) with either X1/Xins or X2/Xins expression greater than two or either X1 or X2 FPKM >3. Genes classified through blastx or panther as receptor tyrosine kinases, protein tyrosine kinases, integrins, GPCRs, or other signaling factors were prioritized. dsRNA was generated through reverse transcription and two rounds of nested PCR with primers indicated (Table S1) and further amplified to add T7 sites for dsRNA production. dsRNA was produced by T7 in vitro transcription, annealing and purification by ethanol precipitation and added to liver as described previously. Animals were fed three times over a week and amputated to remove heads and tails. Head and tail fragments were fixed at 23 days following amputation ('A-score' heads and tails), while trunk fragments were fed again at d10 and amputated to remove heads and tails then regenerating trunk fragments were fixed 12 days later ('B score' trunks). Fixations, fluorescence in

situ hybridizations, and Hoechst stainings to detect *cto* expression were carried out as described previously but in 96-well mesh-bottomed plates (Milipore Multiscreen Plates, MANMN4010) incubating in either Multiscreen receiver plates or rectangular 1-well dishes (VWR 73521–420). Stained animals were mounted and imaged at 40x using a fluorescence dissecting microscope to obtain a view of *cto* expression within the head and also at 25x to obtain a Hoechst-stained view of the entire fragment. A CellProfiler pipeline for automated image analysis was developed to quantify numbers of *cto*+ cells and also to measure area of the fragment as detected by Hoechst staining. Automated cell counting was optimized by adjusting the fixed detection threshold and object size range. Processed images were manually inspected for proper segmentation and scored manually in the event of visible errors in automated counting. Numbers of *cto*+ cells scale with animal length, so we normalized *cto* cell number to the square root of animal area as an approximate of animal length. Log2-fold change of normalized *cto* cell number was computed for each specimen by comparison to the average of control RNAi treatments, and these values for all fragment types (A-score heads and tails, B-score trunks), were binned and plotted in R using ggplot2. T-tests comparing area-normalized *cto* + cell numbers between each RNAi condition and RNAi controls were adjusted for false discovery using the Benjamini-Hochberg method as implemented in R.

## Phylogenetic analysis

Protein sequences were aligned using MUSCLE with default settings (*Chojnacki et al., 2017*). Maximum likelihood analysis was run using PhyML with 100 bootstrap replicates, the WAG model of amino acid substitution, four substitution rate categories, and the proportion of invariable sites estimated from the dataset (*Guindon and Gascuel, 2003*; *Guindon et al., 2010*). We used proteins from *Homo sapiens* (hs), *Mus musculus* (mm), *danio rerio* (dr), *Drosophila melanogaster* (dm), and *Caenorhabditis elegans* (ce).

## RNAi

RNAi was performed by dsRNA feeding. For RNAi, dsRNA was synthesized from in vitro transcription reactions (NxGen, Lucigen). dsRNA corresponding to *Caenorhabditis elegans unc-22*, not present in the planarian genome, served as a negative control. Unless noted otherwise, animals were fed a mixture of liver paste and dsRNA six times in two weeks prior to amputation of heads and tails.

## In Situ hybridization, Immunostaining and qPCR

Animal fixation, bleaching, and in situ hybridization were performed as previously described (*Pearson et al., 2009*; *King and Newmark, 2013*). Briefly, fixation, bleaching, and probe synthesis and hybridization were performed according to Pearson et al. Antibody blocking and tyramide development were performed according to King and Newmark. Digoxigenin- or fluorescein-labeled riboprobes were detected with anti-digoxigenin-HRP (1:2000, Roche/Sigma-Aldrich 11207733910, RRID: AB_514500), anti-fluorescein-HRP (1:2000, Roche/Sigma-Aldrich 11426346910, RRID: AB_840357). Hoechst 33342 (Invitrogen) was used at 1:1000 as a counterstain. Colorimetric (NBT/BCIP) assays were performed as described and detected with anti-digoxigenin-AP (1:4000, Roche/Sigma-Aldrich 11093274910).

For immunostainings, animals were fixed in 4% formaldehyde. Antibodies against mouse anti-tubulin alpha (1:1000 anti-Tubulin Alpha Neomarkers) or rat anti-BrdU (1:1000 Abcam 6326) and detected with goat anti-mouse HRP (1:150 Life Technologies, T20914) or goat anti-rat HRP (1:1000, Jackson ImmunoResearch 112-036-072) respectively. For histone staining, animals were fixed in Carnoy's solution as described (*Umesono et al., 1997*), using tyramide amplification to detect labeling with rabbit anti-phospho-ser10 Histone H3 (1:3000, Cell Signaling D2C8). Rabbit anti-PIWI1 polyclonal antibody (a kind gift of P. Reddien) was used at 1:1000 and detected with goad anti-rabbit HRP (1:150 Life Technologies, T20924).

For qPCR, total animal RNA was collected using Trizol with a tissue homogenizer, reverse transcribed with oligo-dT primers using Superscript II. qPCR was conducted to detect tec-1 mRNA using *tec-1* primers (5'-GTTTTGATGCTAGAATGTTG-3' and 5-TTTGACACACATACTCAAAG-3'), with normalization to a ubiquitously expressed gene gapdh detected with gapdh primers (5'-TGGTATTCAA TTGACCGATACG-3' and 5'-GATCGATTACACGGCAACTG-3') using the delta-Ct method.

## BrdU, TUNEL staining, and Irradiation

For BrdU experiments, the concentration of Montjuic salts were gradually increased to 5x one week before surgery. Animals were treated with 0.0625% N-acetyl cysteine dissolved in 1x Montjuic salts for one minute, washed with 1xMontjuic salts, and incubated in 25 mg/ml BrdU (Sigma) dissolved in 1x salts containing 3% dimethyl sulfoxide for 4 hr either 1 hr before amputation (day 0 soak), or 2 or 4 days post-amputation (day two soak or day four soak respectively). Animals were then maintained in 5x Montjuic salts until fixation on the day indicated in the experiment (*Cowles et al., 2012*; *Zhu and Pearson, 2018*). The BrdU and in situ hybridizations were carried out as previously described above, with all HRP inactivations carried out using formaldehyde (4% in 1xPBSTx for at least 45 min) (*Hill and Petersen, 2018*). Briefly, animals were rehydrated and bleached in 6% hydrogen peroxide in PBSTx for 3–4 hr on a light box. Following FISH protocol described as above, acid hydrolysis was performed in 2N HCl for 45 min, samples were washed with 1xPBS (twice) then 1xPBSTx (four times), and blocked in PBSTB for 6 hr at room temperature. Primary antibody incubation was performed using rat anti-BrdU antibody (1:1000 in PBSTB, Abcam 6326) overnight at room temperature, followed by 6x washes in PBSTB, and overnight incubation in anti-rat HRP secondary antibody (1:1000, Jackson ImmunoResearch 112-036-072). Tyramide development was performed at room temperature for 1 hr (Invitrogen Alexa568-TSA Kit, tyramide at final concentration of 1:150).

Terminal uridine nick-end labeling (TUNEL) was performed as described by *Pellettieri et al. (2010)*, with modifications. Animals were sacrificed in 5% N-acetyl-cysteine in $1 \times$ PBS, fixed in 4% formaldehyde in $1 \times$ PBSTx, and bleached overnight in 6% hydrogen peroxide in $1 \times$ PBSTx. Samples were labeled with DIG-11-dUTP (Roche) by terminal deoxyuridine transferase (TdT) reaction (Thermo) at 37°C for 2 hr, then blocked and incubated overnight in anti-DIG-POD (Roche; 1:2000 in 10% horse serum in $1 \times$ PBSTx) prior to tyramide development (Invitrogen).

To measure neuronal persistence after neoblast ablation, animals were fed and cut as described above. Regenerating trunk fragments were irradiated at 6 days post amputation. Irradiations were carried out in a RS2000 Biological Research Irradiator (Rad Source) at 160 kV over 13'30' for a total, lethal dose of 60 Gy (*Bardeen and Baetjer, 1904*; *Reddien et al., 2005*). Irradiated fragments were fixed at 8, 10, and 12 days post-amputation.

## Image analysis and cell counting

NBT/BCIP-stained animals were imaged with a Leica M210F dissecting microscope and a Leica DFC295, with adjustments to brightness and contrast using Adobe Photoshop. Fluorescent-stained animals were imaged with a Leica TCS SPE confocal microscope or a Leica DM5500B compound microscope.

Unless otherwise noted, cell numbers were counted manually or using Image J's Cell Counter analysis tool (*Schindelin et al., 2012*; *Schneider et al., 2012*). Cell counts were normalized to animal length or brain length using ImageJ or to the square root of the animal's area calculated by Hoechst staining using CellProfiler (*Lamprecht et al., 2007*) as noted. *opsin+* photoreceptor neurons were counted using Imaris x64 7.0.0 (Bitplane AG, Badenerstrasse 682, 8048 Zürich, Switzerland). Surface module was used to define either *opsin+* or *tyrosinase+* volume on each eye and then Spots module was used to count nuclei inside these volumes, as previously described (*Vásquez-Doorman and Petersen, 2016*). Data of cell counts from individual samples used for plotting in each figure panel is presented in *Supplementary file 3*.

For analysis of *cto+* cell size and density, z-stacks of animals stained with a *cto* riboprobe were imaged at 20x on a Leica SPE with one micron slices, and *cto+* cells were segmented in Fiji/ImageJ using the '3D object counter 2.0' plugin after manual thresholding, and the X,Y,Z centroid positions of each cell and cell volume were obtained. Nearest-neighbor analysis was performed by computing the matrix of all pairwise distances between all cell centroids, then for each cell determining the distance between the nearest distinct cell.

## Acknowledgements

We thank members of the Petersen lab for critical comments and P Reddien for the kind gift of the PIWI-1 antibody.

## Additional information

### Funding

| Funder | Grant reference number | Author |
|---|---|---|
| National Institutes of Health | 1R01GM129339 | Christian P Petersen |
| Natural Sciences and Engineering Research Council of Canada | PGSD3-471547-2015 | Nicolle A Bonar |
| National Institutes of Health | R01GM130835 | Christian P Petersen |

The funders had no role in study design, data collection and interpretation, or the decision to submit the work for publication.

### Author contributions

Alexander Karge, Formal analysis, Validation, Investigation; Nicolle A Bonar, Scott Wood, Performed experiments for the screen in Figure 1; Christian P Petersen, Conceptualization, Formal analysis, Supervision, Funding acquisition, Project administration

### Author ORCIDs

Alexander Karge (iD) https://orcid.org/0000-0003-2439-326X
Christian P Petersen (iD) https://orcid.org/0000-0001-7552-6865

### Decision letter and Author response

Decision letter https://doi.org/10.7554/eLife.47293.sa1
Author response https://doi.org/10.7554/eLife.47293.sa2

## Additional files

### Supplementary files

• Supplementary file 1. Information about genes investigated in RNAi screen. Table describes ddv6 contig identifier from Planmine, provisional name, and blastx annotation information obtained from Planmine, as well as primers used for cloning cDNA for each gene.

• Supplementary file 2. RNAi screen data. Table describes the measurement of *cto* cell abundance after inhibition of each gene. Genes are described by their ddv6 contig identifier from Planmine and with a provisional name. Data from analysis of head, trunks, and tail fragments were pooled to obtain an average value of cto cell number normalized to animal area and standard deviations calculated. Log2-normalized values of (average cto cells/area) are additionally presented. Unadjusted and Benjamini-Hochberg-adjusted p-values are shown from t-tests to compare cto/area measurements between each indicated RNAi condition and control RNAi treatment (*C. elegans unc-22*, 'ctrl').

• Supplementary file 3. Data used for plotting figure graphs. Each subpanel or plot is indicated by the name of the corresponding tab in the file. Data of cell counts or normalized cell counts across specimens is indicated.

• Transparent reporting form

### Data availability

All data analyzed during this study are presented in the manuscript and supporting files.

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
