## [Decision Letter]

**Acceptance summary:**

Planarians are organisms possessing extraordinary regenerative abilities, including the complete, functional regeneration of a brain after decapitation. Given the poor capacity of most mammals to regenerate brain tissues, a better understanding of neural regeneration is necessary. Karge and colleagues report on a *tec-1*kinase-mediated negative regulation of planarian neurogenesis. The knockdown of *tec-1* increased the abundance of several types of neurons in the planarian brain during regeneration and homeostasis. Follow-up experiments suggested that *tec-1* functions through facilitating neuronal cell death rather than controlling neuronal differentiation. This work presents an intriguing example of negative regulation of neurogenesis, particularly considering the broad conservation of Tec kinases.

**Decision letter after peer review:**

Thank you for submitting your article "Tec-1 kinase negatively regulates regenerative neurogenesis in planarians" for consideration by *eLife*. Your article has been reviewed by three peer reviewers, and the evaluation has been overseen by a Reviewing Editor and K VijayRaghavan as the Senior Editor. The reviewers have opted to remain anonymous.

The reviewers have discussed the reviews with one another and the Reviewing Editor has drafted this decision to help you prepare a revised submission.

Summary:

In this manuscript, Karge and colleagues report on a *tec-1*kinase-mediated negative regulation of planarian neurogenesis. Through an RNAi screen that includes a focal set of kinases, authors noticed that knockdown of *tec-1* increased the abundance of several types of neurons in the planarian brain during regeneration and homeostasis. Follow-up experiments suggested that *tec-1* functions through facilitating neuronal cell death rather than controlling neuronal differentiation, although the exact molecular mechanism remains unknown. This work presents an intriguing example of negative regulation of neurogenesis, particularly considering the broad conservation of Tec kinases.

Essential revisions:

In order to consider this manuscript further, several major issues in terms of the phenotype and interpretation need to be addressed. These are:

1) The expression of *tec-1* is unclear. While the cell atlas suggests it is broadly expressed and perhaps concentrated in the neuronal progenitors, where is it expressed during regeneration? Does the expression pattern change throughout the regeneration time course? This information is essential, as it should inform the interpretation of phenotypes in terms of whether *tec-1* functions autonomously or non-autonomously. Is *tec-1* co-expressed with any of the neuronal or progenitor types looked at in the paper? Or is it co-expressed with macrophage/phagocytic markers? If FISH is challenging in whole-mount, it is worth trying in sections and dissociated cells.

2) Control and *tec-1*RNAi animals were compared in terms of differences in differentiation 8 days upon amputation. A BrdU pulse was delivered on day 2 to capture to trace the progeny of cells that were in S phase during the pulse (the duration of this pulse is not specified but should be). The result of this experiment is clear – no significant difference in the number of BrdU^+^ cells in the ppl+ cell population (which was affected in *tec-1*RNAi much like *cto+* cells). While this work supports the conclusion that *tec-1*RNAi does not affect differentiation of cells in S-phase on day 2, it does not rule out differences in differentiation that may occur prior to or later than this time point. The numbers of progenitors marked by coe, pax6, and pitx were only compared 8 days post amputation, which also does not eliminate the idea that differentiation might be affected by *tec-1* at other time points. Is it known that commitment to differentiated fates only begins on day 2 and that differentiation commitment later than day 2 does not contribute to fully differentiated cell types in a significant way? Or that differentiation rates are constant over time as the worms regenerate? Could the original observation that *tec-1*RNAi animals show larger numbers of *cto+* cells later in regeneration be explained by really early (before day 2) differences?

3) Does *tec-1* function similarly during regeneration and homeostasis? In Figure 4A, the difference between control and *tec-1*RNAi increases over time, suggesting more neurons should be removed as regeneration progresses. But the system has to reach a steady state eventually, but there is no data showing this. How and when might the steady state be reached in this case? Is it possible that *tec-1*RNAi animals just have a delayed peak of neuronal deaths? In this case, the steady states may be similar between control and RNAi. The comparison of neurons between control and RNAi conditions should be performed after the system reaches a steady state.

4) In the *tec-1*RNAi the numbers of many neuronal cell types go up, in particular in the brain region, but the size of the brain doesn't change. It is clearly visible in the staining that the density of the evaluated neuronal cell types is increased. But how does this work? Is the general cell density in the brain region higher? Are the cells smaller? Or are there compensatory losses of other cell types in the region? Could the detected loss in glial cells be sufficient to compensate?

5) The authors state that there are no changes in polarity or global patterning genes, but Figure 2B shows a clear increase in the number of notum-expressing cells, and Figure 3B clearly shows disorganized photoreceptors. This suggests that there are definitely some – possibly subtle – changes in anterior organization. It would be very interesting to understand whether the change in notum levels could induce this change in responsiveness to apoptotic signals. In addition, there have been some indications that apoptosis is not constant along the A-P axis (e.g. Peiris et al., 2016), and it would be worth taking into consideration whether this might explain some of the observed phenotype.

6) Is there an anterior bias in the detected neuronal cell increase? Most of the neuronal cell types are evaluated in the anterior region. Only the bottom panel in Figure 3A shows a change in a more posterior region. Are the effects mostly detected in the anterior, or are neuronal cell increases in the posterior just as strong?

7) Is it possible that *tec-1*RNAi causes a prolonged differentiation? There is a caveat in terms of Figure 4C showing *tec-1*'s disinvolvement with rate of neuronal differentiation. The regenerating tail fragments were examined at day 8 post-amputation, at which point Figure 4A suggests has only a slight increase in neuronal abundance and thus is statistically unlikely to be noticeable at this stage within a smaller subset of neuronal types. A different time point may address the possibility of delayed differentiation.

8) The phenotype should be characterized and discussed with greater depth. Do the ratios between different neuronal types change? This can be assayed with double FISH. If the ratios have changed, how should we understand the activity of *tec-1* in different neuronal populations? Are there any neuronal types within the CNS or PNS that reduce in numbers or are not affected after RNAi? Neuronal types unaffected by *tec-1* may bring important insights and hypotheses. Readers would appreciate the authors' insights to these questions. If all neuronal types increase in numbers within a constant brain size, it might suggest that the neurons must reduce their sizes as they experience a cellular overcrowding effect. Is this effect measurable (e.g., through nucleus-nucleus distance)? Are the changes in number density and cell size consistent? The authors have shown that glial cell number mildly decreases, which is very interesting, but is this change enough to compensate the increase of all neurons? How should we interpret this result – is it explained by the hypothesis of a distinct cell population killing neurons, or might this have any basis in a potential differentiation pathway?

9) The choice of timing for the irradiation experiment needs to be explained as well. This experiment shows that in the absence of new cell production, *tec-1*RNAi animals continue to expand numbers of *cto+* cells, whereas control RNAi animals show a decrease in *cto+* cells at day 12 relative to day 10. However, Figure 4B showed a statistically significant different in *cto+* cells between control and *tec-1* RNAi animals on day 10 (which was also detectable as significant on day 8). If differentiation is not a source of *cto+* differences between control and *tec-1*RNAi animals, one would expect the differences to be the same upon elimination of the contribution of new cells (at least at day 8). What explains the non-significant differences at day 8 and 10 in Figure 5B? Do the results argue that both differentiation and cell survival are affected?

10) The authors report that Tec kinases are known to control inflammation and recovery following traumatic brain injury by regulating macrophages, which does not directly imply that Tec kinase homologs regulate neuronal cell numbers via controlling their survival. Therefore, the next statement (“We suggest that Tec kinases could regulate specific and perhaps conserved functions in adult neurogenesis process in control of neuron abundance”) is not well-supported. There are no data presented that enable the proposal of a putative conserved function for *tec-1*. The authors' finding that planarian regeneration involves overproduction of neural cells followed by cell death bears overall similarity to the mechanisms of vertebrate development where cell death plays a major role in shaping the nervous system. However, given the lack of demonstrated similarity of function of the ortholog of *tec-1* in neural development in vertebrates, the claims of *tec-1* being a promising new target for therapeutic interventions are premature.

11) Finally, the finding of increased density of glial cells in *tec-1*RNAi is interesting, and the authors should discuss if this could be suggestive of *tec-1* being involved in a fate choice – could glial cells and *cto+* cells be differentiating from a shared progenitor population?

---

## [Author Response]

Essential revisions:In order to consider this manuscript further, several major issues in terms of the phenotype and interpretation need to be addressed. These are:1) The expression of tec-1 is unclear. While the cell atlas suggests it is broadly expressed and perhaps concentrated in the neuronal progenitors, where is it expressed during regeneration? Does the expression pattern change throughout the regeneration time course? This information is essential, as it should inform the interpretation of phenotypes in terms of whether tec-1 functions autonomously or non-autonomously. Is tec-1 co-expressed with any of the neuronal or progenitor types looked at in the paper? Or is it co-expressed with macrophage/phagocytic markers? If FISH is challenging in whole-mount, it is worth trying in sections and dissociated cells.

We appreciate this suggestion and now show FISH images of *tec-1* expression in whole animals. Indeed, in confirmation of the scRNAseq data, we find *tec-1* FISH signal to be broadly distributed, with co-expression analysis confirming expression in several different neurons, *piwi-1+* cells, and glia (Figure 4—figure supplement 3). We also examine overall *tec-1* expression during regeneration (Figure 4—figure supplement 1) and find it is also broadly expressed during regeneration with no obvious injury-induced expression behavior. The broadness of expression implies that we cannot at present rule out the possibility of *tec-1* acting within either neural progenitors, differentiated neurons, or some other cell type to exert its function on neuron numbers, and we clarify this interpretation in the text (Discussion paragraph three).

2) Control and tec-1 RNAi animals were compared in terms of differences in differentiation 8 days upon amputation. A BrdU pulse was delivered on day 2 to capture to trace the progeny of cells that were in S phase during the pulse (the duration of this pulse is not specified but should be). The result of this experiment is clear – no significant difference in the number of BrdU^+^ cells in the ppl+ cell population (which was affected in tec-1 RNAi much like cto+ cells). While this work supports the conclusion that tec-1 RNAi does not affect differentiation of cells in S-phase on day 2, it does not rule out differences in differentiation that may occur prior to or later than this time point. The numbers of progenitors marked by coe, pax6, and pitx were only compared 8 days post amputation, which also does not eliminate the idea that differentiation might be affected by tec-1 at other time points. Is it known that commitment to differentiated fates only begins on day 2 and that differentiation commitment later than day 2 does not contribute to fully differentiated cell types in a significant way? Or that differentiation rates are constant over time as the worms regenerate? Could the original observation that tec-1 RNAi animals show larger numbers of cto+ cells later in regeneration be explained by really early (before day 2) differences?

We have repeated the BrdU experiment by adding multiple time points for BrdU soaking and animal fixation during regeneration in Figure 4B (all combinations of soaking at day 0, 2, 4 and fixing at regeneration day 6, 8, 10, 12). These experiments did not find any differences between numbers of BrdU+*ppl-1*+ cells across all timepoints of regeneration or soaking tested. To support these conclusions, we also conducted an expanded neural progenitor counting experiments across a large range of timepoints in regeneration (Figure 4C), and confirm that we could not detect any excess rates of differentiation in *tec-1*RNAi animals.

3) Does tec-1 function similarly during regeneration and homeostasis? In Figure 4A, the difference between control and tec-1 RNAi increases over time, suggesting more neurons should be removed as regeneration progresses. But the system has to reach a steady state eventually, but there is no data showing this. How and when might the steady state be reached in this case? Is it possible that tec-1 RNAi animals just have a delayed peak of neuronal deaths? In this case, the steady states may be similar between control and RNAi. The comparison of neurons between control and RNAi conditions should be performed after the system reaches a steady state.

We now include a longer time course showing that the steady state is reached by day 14 and the phenotype persisted through at least 28 days of regeneration (Figure 4—figure supplement 2). Based on observing similar overall phenotypes in regeneration or homeostasis assays, we suggest *tec-1* has a common role in negatively regulating neurogenesis, with each particular assay influencing the kinetics of response.

4) In the tec-1 RNAi the numbers of many neuronal cell types go up, in particular in the brain region, but the size of the brain doesn't change. It is clearly visible in the staining that the density of the evaluated neuronal cell types is increased. But how does this work? Is the general cell density in the brain region higher? Are the cells smaller? Or are there compensatory losses of other cell types in the region? Could the detected loss in glial cells be sufficient to compensate?

To consider this, we performed 3D segmentation of *cintillo* cells from confocal stacks and made measurements of cell volume and the nearest-neighbor distances for each cell. This approach found that in *tec-1 RNAi, cintillo* cells were more numerous and slightly smaller, and also that they are more densely packed with respect to each other (Figure 2—figure supplement 2). We suggest that increased density along with reduced size accounts for how more neurons physically occupy apparently similar brain volumes in *tec-1*RNAi.

5) The authors state that there are no changes in polarity or global patterning genes, but Figure 2B shows a clear increase in the number of notum-expressing cells, and Figure 3B clearly shows disorganized photoreceptors. This suggests that there are definitely some – possibly subtle – changes in anterior organization. It would be very interesting to understand whether the change in notum levels could induce this change in responsiveness to apoptotic signals. In addition, there have been some indications that apoptosis is not constant along the A-P axis (e.g. Peiris et al., 2016), and it would be worth taking into consideration whether this might explain some of the observed phenotype.

To address this, we used FISH to more carefully detect and pinpoint the excess *notum+* cells. These excess cells were found to be within a previously described set of *notum+chat+* cells of the brain (Figure 2—figure supplement 1), which we interpret to be consistent with the supported role of *tec-1* to limit numbers of multiple types of neurons. We suggest based on our analysis that *tec-1* likely acts in an independent process from other known head patterning genes (such as *notum, wnt11-6/wntA, ndk*, etc), because *tec-1* inhibition primarily influences neuron cell density versus head regionalization.

We appreciate the suggestion to consider whether *tec-1* could be part of the previously characterized A-P axis apoptotic pathway. To test this, we searched the recently published scRNAseq planarian cell atlas to find a neuron subtype that was broadly distributed but disperse enough that total animal cell quantifications could be carried out confidently. Of the tested neuron subtypes, we found one marker (*dd2223*, labeling nonciliated neuron cluster#32 identified by Fincher et al., 2018) that was strongly sensitive to *tec-1* inhibition. In *tec-1*RNAi animals, we found an equivalent increase in *dd2223+* neuron cells in anterior versus posterior body regions (Figure 3—figure supplement 2). Therefore, our findings suggest *tec-1* can exert its influence independent of A-P axis position, so we suggest it is likely part of a distinct pathway from factors specifically influencing anterior versus posterior cell survival.

6) Is there an anterior bias in the detected neuronal cell increase? Most of the neuronal cell types are evaluated in the anterior region. Only the bottom panel in Figure 3A shows a change in a more posterior region. Are the effects mostly detected in the anterior, or are neuronal cell increases in the posterior just as strong?

Similar to above, to test this, we searched the recently published scRNAseq planarian cell atlas to find a neuron subtype that was broadly distributed but disperse enough that total animal cell quantifications could be carried out confidently. Of the tested neuron subtypes, we found one marker (*dd2223*, labeling nonciliated neuron cluster#32 identified by Fincher et al., 2018) that was strongly sensitive to *tec-1* inhibition. In *tec-1*RNAi animals, we found an equivalent increase in *dd2223+* neuron cells in anterior versus posterior body regions (Figure 3—figure supplement 2). Therefore, *tec-1* can exert its influence independent of AP axis position.

7) Is it possible that tec-1 RNAi causes a prolonged differentiation? There is a caveat in terms of Figure 4C showing tec-1's disinvolvement with rate of neuronal differentiation. The regenerating tail fragments were examined at day 8 post-amputation, at which point Figure 4A suggests has only a slight increase in neuronal abundance and thus is statistically unlikely to be noticeable at this stage within a smaller subset of neuronal types. A different time point may address the possibility of delayed differentiation.

We tested the possibility of delayed differentiation through an extended set of measurements over time (2, 4, 8, 12 days head regeneration) and for 3 different neural progenitors during regeneration. In all three cases (*coe, pax6A, pitx*), we found that both control and *tec-1*RNAi animals undergo a similar reduction of progenitor numbers during regeneration. There was no significant difference in progenitor numbers between control and *tec-1*RNAi animals at all timepoints throughout regeneration (day 2-day 12) (Figure 4C). Therefore, we do not find evidence for *tec-1* inhibition prolongs a period of higher differentiation in regeneration.

8) The phenotype should be characterized and discussed with greater depth. Do the ratios between different neuronal types change? This can be assayed with double FISH. If the ratios have changed, how should we understand the activity of tec-1 in different neuronal populations? Are there any neuronal types within the CNS or PNS that reduce in numbers or are not affected after RNAi? Neuronal types unaffected by tec-1 may bring important insights and hypotheses. Readers would appreciate the authors' insights to these questions. If all neuronal types increase in numbers within a constant brain size, it might suggest that the neurons must reduce their sizes as they experience a cellular overcrowding effect. Is this effect measurable (e.g., through nucleus-nucleus distance)? Are the changes in number density and cell size consistent? The authors have shown that glial cell number mildly decreases, which is very interesting, but is this change enough to compensate the increase of all neurons? How should we interpret this result – is it explained by the hypothesis of a distinct cell population killing neurons, or might this have any basis in a potential differentiation pathway?

Using double-FISH, we now include experiments to examine the ratio of *cto*+ neurons to other neuron types in Figure 3—figure supplement 3. These revealed no alteration of cto:gad or cto:ppl1 cell ratios after *tec-1*RNAi, suggesting *tec-1*concordantly affects the abundance of multiple neuron types.

To expand the analysis of the phenotype we examined five additional neuron subtypes identified in the recent scRNAseq cell atlas for planarians. Though planarians may have ~50 different types of neurons, the effect size of the *tec-1*RNAi phenotype led us to focus on cell types for which total body enumeration could be performed at a high degree of precision. Of the 5 genes we found to meet this criteria, 4 underwent tec-1 dependent increases (Figure 3A and Figure 3—figure supplement 2): *dd2223, dd3733, spp-4, dd17258*) and the other did not change in abundance after *tec-1*RNAi (*dd2723*). Altogether, our analysis now finds 10 of 13 neuron types increased after *tec-1*RNAi, and 3 of 13 did not change (note we found none that decreased in number). Those unaffected by *tec-1* inhibition did not have any known discernable commonality in terms of location or distribution (opsin+ photoreceptor neurons, *ppl1+* pharyngeal neurons, *dd2723+* CNS cells of the brain, ventral nerve cords, and pharynx). These findings, in combination with the broad expression of *tec-1*, prevented us from identifying a best hypothesis at this time for why some neurons are affected by *tec-1* and some are not.

We appreciate the suggestion to determine how extra neurons in *tec-1(RNAi)* animals physically pack in space, given for example that overall brain size is not affected. To address this, we performed 3D segmentation on *cintillo*-stained animals and computed volume and nearest neighbor distances between the cells (Figure 2—figure supplement 1). This approach found that *tec-1* inhibition indeed results in overcrowding of *cto+* neurons, leading to higher neuron densities and smaller volumes of *cto+* cells. Perhaps as extra neurons grow, some process constrains overall tissue volume, leading to these cells attaining reduced sizes. Based on the separation of regions occupied by glia and the affected neuron cell types, we believe the crowding effect to be a separate phenomenon than loss of glial abundance or size. In combination with additional experiments showing lack of support for increased differentiation of neurons (BrdU staining, progenitor counting), our analysis suggests *tec-1* is unlikely to function in a switch to control glial-vs-neuron differentiation.

9) The choice of timing for the irradiation experiment needs to be explained as well. This experiment shows that in the absence of new cell production, tec-1 RNAi animals continue to expand numbers of cto+ cells, whereas control RNAi animals show a decrease in cto+ cells at day 12 relative to day 10. However, Figure 4B showed a statistically significant different in cto+ cells between control and tec-1 RNAi animals on day 10 (which was also detectable as significant on day 8). If differentiation is not a source of cto+ differences between control and tec-1 RNAi animals, one would expect the differences to be the same upon elimination of the contribution of new cells (at least at day 8). What explains the non-significant differences at day 8 and 10 in Figure 5B? Do the results argue that both differentiation and cell survival are affected?

We chose this timing in order to attempt to generate a pulse of new neurons whose behavior could be assessed over time. In contexts of head regeneration, the exact timing of the onset of the *tec-1* excess *cto+* cell phenotype was somewhat variable across experiments, dependent on feeding schedule and fragment type (note that Figure 4A analyzed tail fragments while Figure 5B analyzed trunk fragments). For the irradiation experiment, we chose trunk fragments because they can survive for longer as compared to irradiated tail fragments. In those fragments, we never observed the excess *cto+* phenotype at d8 but only observed it emerge at later times – by day 12 in head regeneration, the phenotype was always observed. We repeated the experiment in Figure 5B three times and observed in each case the *tec-1*RNAi phenotype in irradiated animals by day 12. Our interpretation of the BrdU labeling and progenitor counting experiments is that we were unable to observe any effect of *tec-1* inhibition on differentiation.

10) The authors report that Tec kinases are known to control inflammation and recovery following traumatic brain injury by regulating macrophages, which does not directly imply that Tec kinase homologs regulate neuronal cell numbers via controlling their survival. Therefore, the next statement (“We suggest that Tec kinases could regulate specific and perhaps conserved functions in adult neurogenesis process in control of neuron abundance”) is not well-supported. There are no data presented that enable the proposal of a putative conserved function for tec-1. The authors' finding that planarian regeneration involves overproduction of neural cells followed by cell death bears overall similarity to the mechanisms of vertebrate development where cell death plays a major role in shaping the nervous system. However, given the lack of demonstrated similarity of function of the ortholog of tec-1 in neural development in vertebrates, the claims of tec-1 being a promising new target for therapeutic interventions are premature.

We appreciate this suggestion to the presentation. We clarified our reference to the study noted above, that it found an increase in brain cell survival (ie, reduced staining of cell death markers) in an ischemic brain injury model after pharmacological inhibition of the Tec-family kinase BTK in mammals (Discussion paragraph five).

We also modified the text suggesting that our study necessarily implies the existence of a conserved activity of *tec-1* to suppress neurogenesis. Given that some components of neurogenesis are broadly conserved, we think it is worthwhile to raise the idea that future studies could address this possibility. In addition, we now tone down statements of the concept for a potential use for *tec-1* inhibition to enhance neurogenesis (given additional study). We hope our revised text makes clear that we do not present evidence for a conserved function in this study:

Abstract: “In vertebrates, the Tec kinase family has been studied extensively for roles in immune function, and our results identify a novel role for *tec-1* as negative regulator of planarian adult neurogenesis.”

Discussion paragraph five: “If the ability of Tec Kinases to limit neuron production ultimately proves to be conserved, they would be potential targets for enhancing neural repair in other species.”

Discussion paragraph four: “It is possible that Tec kinases could regulate conserved functions in adult neurogenesis processes in control of neuron abundance.”

11) Finally, the finding of increased density of glial cells in tec-1 RNAi is interesting, and the authors should discuss if this could be suggestive of tec-1 being involved in a fate choice – could glial cells and cto+ cells be differentiating from a shared progenitor population?

We appreciate this suggestion, as it was indeed our initial hypothesis leading us to examine the glia in *tec-1*RNAi. We have added text discussing the hypothesis of a *tec-1* functioning in a neuron-glial differentiation switch, but also how our experiments would suggest against this model because of finding no evidence for *tec-1* inhibition to increase rates of neuron differentiation (subsection “*tec-1* suppresses neuronal cell number by regulating cell survival”).